# Glutathione Deficiency during Early Postnatal Development Causes Schizophrenia-Like Symptoms and a Reduction in BDNF Levels in the Cortex and Hippocampus of Adult Sprague–Dawley Rats

**DOI:** 10.3390/ijms22126171

**Published:** 2021-06-08

**Authors:** Marta Anna Lech, Monika Leśkiewicz, Kinga Kamińska, Zofia Rogóż, Elżbieta Lorenc-Koci

**Affiliations:** 1Department of Pharmacology, Maj Institute of Pharmacology, Polish Academy of Sciences, 12 Smętna Street, 31-343 Kraków, Poland; hereta@if-pan.krakow.pl (M.A.L.); k.kamin@if-pan.krakow.pl (K.K.); rogoz@if-pan.krakow.pl (Z.R.); 2Department of Experimental Neuroendocrinology, Maj Institute of Pharmacology, Polish Academy of Sciences, 12 Smętna Street, 31-343 Kraków, Poland; leskiew@if-pan.krakow.pl; 3Department of Neuro-Psychopharmacology, Maj Institute of Pharmacology, Polish Academy of Sciences, 12 Smętna Street, 31-343 Kraków, Poland

**Keywords:** neurodevelopmental model of schizophrenia, schizophrenia-like symptoms, levels of BDNF mRNA and its protein, effect of amphetamine

## Abstract

Growing body of evidence points to dysregulation of redox status in the brain as an important factor in the pathogenesis of schizophrenia. The aim of our study was to evaluate the effects of l-buthionine-(*S*,*R*)-sulfoximine (BSO), a glutathione (GSH) synthesis inhibitor, and 1-[2-Bis(4-fluorophenyl)methoxy]ethyl]-4-(3-phenylpropyl)piperazine dihydrochloride (GBR 12909), a dopamine reuptake inhibitor, given alone or in combination, to Sprague–Dawley pups during early postnatal development (p5–p16), on the time course of the onset of schizophrenia-like behaviors, and on the expression of brain-derived neurotrophic factor (BDNF) mRNA and its protein in the prefrontal cortex (PFC) and hippocampus (HIP) during adulthood. BSO administered alone decreased the levels of BDNF mRNA and its protein both in the PFC and HIP. Treatment with the combination of BSO + GBR 12909 also decreased BDNF mRNA and its protein in the PFC, but in the HIP, only the level of BDNF protein was decreased. Schizophrenia-like behaviors in rats were assessed at three time points of adolescence (p30, p42–p44, p60–p62) and in early adulthood (p90–p92) using the social interaction test, novel object recognition test, and open field test. Social and cognitive deficits first appeared in the middle adolescence stage and continued to occur into adulthood, both in rats treated with BSO alone or with the BSO + GBR 12909 combination. Behavior corresponding to positive symptoms in humans occurred in the middle adolescence period, only in rats treated with BSO + GBR 12909. Only in the latter group, amphetamine exacerbated the existing positive symptoms in adulthood. Our data show that rats receiving the BSO + GBR 12909 combination in the early postnatal life reproduced virtually all symptoms observed in patients with schizophrenia and, therefore, can be considered a valuable neurodevelopmental model of this disease.

## 1. Introduction

Schizophrenia is a severe chronic mental illness affecting approximately 1% of the world population [1] and characterized by three broad categories of symptoms, namely positive symptoms (delusions, hallucinations, thought disorder, and incoherence) and negative symptoms (lack of motivation and deficits in social function) as well as cognitive impairment (decline of working memory, executive function, learning, long-term memory, visual/auditory perception, and attention) [2,3,4]. Positive symptoms are the most striking feature of the disease, but cognitive deficits, typically present before the onset of psychosis [5], are critical determinants of patients’ quality of life and their daily functioning [6,7,8]. The etiology of schizophrenia still remains poorly understood, but according to the dominant hypothesis, it is increasingly recognized as a disease with an important neurodevelopmental component contributing to structural and functional changes in the brain. It is believed that the aberrant neurodevelopmental processes are induced by multiple interactions between the genetic and environmental factors [9] that occur during embryonic or early postnatal development. Detrimental effects of these interactions result in the emergence of cognitive deficits and other symptoms of schizophrenia in adolescence and/or early adulthood [3,10,11,12,13].

Although the pathomechanism of schizophrenia is not fully explored, an increasing number of studies indicate oxidative stress, the impaired redox status of cells, and epigenetic regulation disorders as key factors contributing to the pathophysiology of this disease [14,15,16,17,18,19,20,21]. Referring to the impaired redox status in schizophrenia, it has been shown that the level of its main regulator and strong antioxidant, glutathione (GSH), was clearly reduced in the cerebrospinal fluid and medial frontal cortex of drug-naïve schizophrenic patients [22] and also in the post-mortem striatum [23] and the prefrontal cortex of those previously treated with antipsychotic drugs [24]. Decreases in GSH levels were also found in erythrocytes [25,26,27] and in plasma [26,28,29] of antipsychotic-free schizophrenia patients as well as in those chronically medicated with these drugs. Impairment of GSH synthesis in some patients with schizophrenia has been shown to be linked with polymorphisms in genes encoding both catalytic and modifier subunits of γ-glutamate-cysteine ligase (GCL) [30,31,32], a key enzyme catalyzing the first stage of the two-step reaction of GSH formation. Furthermore, a significant negative correlation was found between the brain GSH levels and the severity of negative symptoms in schizophrenia patients [33].

The occurrence of schizophrenia-like symptoms [34,35,36,37,38,39], reminiscent of those observed in patients, has been described in rodents in which the GSH deficit was induced by specific compounds that reduce GSH concentration [40,41]. Behavioral consequences of brain GSH deficit, induced by chronic administration of the GCL inhibitor, l-buthionine-(*S*,*R*)-sulfoximine (BSO), in combination with a dopamine (DA) reuptake inhibitor, the compound 1-[2-Bis(4-fluorophenyl)methoxy]ethyl]-4-(3-phenylpropyl)piperazine dihydrochloride (GBR 12909), during early postnatal development (p5–p16) were studied for the first time in Osteogenic disorder Shionogi (ODS) mutant rats, which, like humans, cannot synthesize ascorbic acid [37,38]. In these studies, it was demonstrated that such combined treatment with BSO + GBR 12909 in early postnatal life caused the long-term schizophrenia-like memory deficits, assessed in the novel object recognition test (NOR) in adulthood. Correspondingly, treatment of ODS and Wistar rats during early postnatal life with BSO alone evoked impairments of some cognitive functions under conditions, when learning and discrimination tasks aimed at assessing spatial working memory performance (homing board, radial maze) were tested in the presence of controlled olfactory cues in adulthood [39].

In our recently published study by Górny et al. [42] conducted on Sprague–Dawley rats chronically treated with BSO alone or the BSO + GBR 12909 combination on postnatal days p5 to p16, cognitive deficits, as well as previously unidentified deficits in social behavior, were found in adulthood (p90–p91). However, only in rats treated with the BSO + GBR 12909 combination, the elevated values of exploratory behavior parameters (time of walking, ambulation, peeping, and rearing), which correspond to the positive symptoms in schizophrenia patients, were observed during the 3-min open field test (OFT) in adulthood (p92). The effects described above clearly indicate that inhibition of both GSH synthesis and dopamine reuptake in the early postnatal life leads to the manifestation of behaviors resembling positive symptoms of schizophrenia in adulthood, while inhibition of GSH synthesis alone at this stage of development is sufficient for the disclosure of deficits in the social behavior and cognitive functions.

The aim of the present study was to further characterize this neurodevelopmental rat model of schizophrenia and to establish at what time point after administration of BSO alone or in combination with GBR 12909 to Sprague–Dawley pups, the first episodes of schizophrenia-like behavior appear. To address these issues, schizophrenia-like behaviors, corresponding to negative and positive symptoms as well as to cognitive deficits, were assessed using behavioral tests (social interaction test, SIT; NOR; OFT) at three time points of adolescence (p30, p42–p44, p60–p62) as well as in adulthood (p90–p92). In schizophrenia patients, it has been shown that the increased release of DA in the striatum induced by amphetamine (AMF), as determined by means of position-emission tomography, was associated with the appearance or exacerbation of positive symptoms [43,44,45]. Therefore, to investigate whether Sprague–Dawley rats given the BSO + GBR 12909 combination during the early postnatal life respond behaviorally in a similar way as patients with schizophrenia, we measured the AMF-stimulated locomotor activity for 30 min using actometers in adulthood, as an indicator of exacerbation of positive symptoms. In addition, to investigate whether the inhibition of GSH synthesis and dopamine reuptake in early postnatal life could affect the brain-derived neurotrophic factor (BDNF), we examined both BDNF mRNA and protein levels in the prefrontal cortex (PFC) and the hippocampus (HIP) in 93-day-old rats after completion of the last series of behavioral tests. In the brain, BDNF is an important signaling molecule responsible for neuronal growth, maturation of synapses during development, and synaptic plasticity [46,47,48], which is essential for learning and memory processes [49,50]. It is worth noting that administration of BSO in the early postnatal days (p5–p16) coincided with the developmental switch in the action of GABA, through GABA_A_ receptors, from excitatory to inhibitory [51], as well as with a peak of synaptogenesis and the formation of adult neuronal networks [52]. In the rat brain, the critical period of increased excitability that takes place during the second postnatal week is within the temporal window of excitatory action of GABA [53,54]. Interestingly, a growing body of experimental evidence indicates that during this period, the excitatory action of GABA participates in the BDNF-mediated signaling and induction of synaptic plasticity in the developing hippocampus [55,56,57,58]. On the other hand, it has been demonstrated that GABA_A_ receptor agonists inhibit BDNF expression [59,60,61]. In line with these facts, the formation of normal neuronal connections during early brain development depends on a precise balance between excitation and inhibition. Therefore, it is reasonable to assume that even a small impairment of this process may lead to developmental abnormalities. However, till now, it has not been explored whether inhibition of GSH synthesis and consequent impairment of the redox status of brain cells early in postnatal life [62], when GSH concentration in this tissue is the highest [63], might interfere with the excitatory action of GABA during this time, causing changes in BDNF expression in the PFC and HIP visible in adulthood. Hence, it seems that the determination of BDNF levels in the groups of rats receiving BSO may help answer this question and explain the link between the GSH deficit in early postnatal life and the occurrence of negative symptoms and memory deficits in adulthood. We hope that the obtained results bring a new quality to the search for the most adequate animal model of schizophrenia.

## 2. Results

### 2.1. The Impact of Chronic Administration of BSO and GBR 12909 during the Early Postnatal Life on the Development of Social Deficits in Adolescence and Adulthood

Social behavior was assessed by means of two parameters, i.e., the total time spent by two rats in social behavior and the number of these interactions. Figure 1A,B shows the time-dependent changes in the expression of these two parameters during adolescence and adulthood of rats that were treated chronically with BSO and GBR 12909 alone or jointly during early postnatal life (p5–p16).

A two-way ANOVA performed for the total time spent in social interactions revealed an overall treatment effect of BSO (for p42—F_(1,28)_ = 36.327, *p* < 0.001; for p60—F_(1,28)_ = 70.315, *p* < 0.001; for p92—F_(1,28)_ = 30.258, *p* < 0.001) but a lack of GBR 12909 treatment effect and no interaction between these two model substances in any of the studied time points (Figure 1A). Similarly, a two-way ANOVA carried out for the number of social interactions showed only a significant treatment effect of BSO (for p42—F_(1,28)_ = 20.747, *p* < 0.001; for p60—F_(1,28)_ = 38.099, *p* < 0.001; for p92—F_(1,28)_ = 17.648, *p* < 0.001) on this parameter at three out of four studied time points (Figure 1B).

Post hoc comparisons of the effects of model compounds on the studied parameters showed that BSO administered alone or in combination with GBR 12909 shortened the total time spent in social interactions (Figure 1A) as well as decreased the number of these interactions (Figure 1B) for the first time in rats that reached the age of 42 days. These effects were still present in late adolescence (at p60) and in adulthood (at p90). In contrast to BSO, administration of GBR 12909 alone during early postnatal life did not evoke changes in the measured parameters of social behavior at any of the studied time points of adolescence and adulthood.

### 2.2. The Impact of Chronic Administration of BSO and GBR 12909 during the Early Postnatal Life on the Development of Cognitive Deficits in Adolescence and Adulthood

The NOR test serving to evaluate cognitive impairments in rodents was performed on the next day after SIT in all studied groups at each time point (Figure 2).

During the acquisition trial (session T1), all rats representing particular groups spent equal time exploring two identical objects (Figure 2A1,A2). In the retention trial (session T2) adolescent control 43- and 61-day-old, as well as adult 91-day-old control rats, but not 31-day-old animals, explored the novel object significantly longer than the familiar one. (Figure 2A2–D2). Interestingly, in 43-day-old rats that were treated with GBR 12909 alone in early postnatal life (p5–p16), exploration of the familiar object was markedly less intensive than the novel one (Figure 2B2). However, in 61-day-old rats treated with GBR 12909, an increasing tendency toward the exploration of the novel object was observed, while in 91-day-old ones, the time of exploration of the novel object was significantly longer than the familiar one, like in controls (Figure 2C2,D2).

Two-way ANOVA performed in groups of adolescent 31-day-old rats demonstrated neither the effects of the tested model compounds nor their combination on the values of the recognition indexes (Figure 2A3). However, this analysis performed for the recognition index in adolescent 42-day-old rats (Figure 2B3) revealed a significant treatment effect of GBR 12909 (F_(1,36)_ = 109.69, *p* < 0.001) and an interaction of BSO × GBR 12909 (F_(1,36)_ = 79.341, *p* < 0.001), but a lack of treatment effect of BSO alone (F_(1,36)_ = 0.989, NS). However, in adolescent 61-day-old rats a two-way ANOVA showed both significant treatment effect of BSO (F_(1,36)_ = 14.797, *p* < 0.001) and GBR 12909 (F_(1,36)_ = 5.232, *p* < 0.05) but no interaction between these model compounds (F_(1,36)_ = 0.041, NS) (Figure 2C3). Furthermore, in adult 91-day-old rats only treatment effect of BSO (F_(1,36)_ = 71.530, *p* < 0.001) was observed (Figure 2D3). Post hoc comparison showed that values of the recognition indexes in group of rats treated with BSO alone or in combination with GBR 12909 were significantly decreased compared to the value of this parameter in the control group in adolescent (p43, p61) and in adult rats (p91).

### 2.3. The Impact of Chronic Administration of BSO and GBR 12909 during the Early Postnatal Life on the Manifestation of Positive Lescence and Adulthood

On the 32nd, 44th, 62nd, and 92nd days of postnatal life, the exploratory activity in the open field test (time of walking, ambulation, peeping, and rearing) was determined in all studied groups of rats as a measure of positive symptoms (Figure 3A–C).

A two-way ANOVA performed for the time of walking revealed a significant treatment effect of BSO in middle and late adolescence and in adulthood (for p44 (F_(1,36)_ = 4.563, *p* < 0.05); for p62 (F_(1,36)_ = 39.160, *p* < 0.0001); for p92 (F_(1,36)_ = 52.678, *p* < 0.0001)) but not in early adolescence (for p32 (F_(1,36)_ = 0.277, NS). This analysis also demonstrated a significant treatment effect of GBR 12909 on this parameter, at all time points (for p32 (F_(1,36)_ = 4.429, *p* < 0.05); for p44 (F_(1,36)_ = 5.003, *p* < 0.05); for p62 (F_(1,36)_ = 6.006, *p* < 0.05)) except the adult 92-day-old rats (F_(1,36)_ = 0.277, NS). Furthermore, a significant interaction of BSO and GBR 12909 regarding the time of walking was found at all studied time points (for p32 (F_(1,36)_ = 4.429, *p* < 0.05); for p44 (F_(1,36)_ = 18.904, *p* < 0.001); for p62 (F_(1,36)_ = 83.917, *p* < 0.001); for p92 (F_(1,36)_ = 67.522, *p* < 0.0001)). Post hoc comparisons showed that the combined treatment with these model compounds resulted in the extension of walking time in the adolescent 44-, 62- and adult 92-day-old rats when compared to rats treated chronically with vehicle (control), BSO or GBR12909 alone during early postnatal life (Figure 3A). Such effect of the combined treatment was not observed in adolescent, 32-days old rats. On the other hand, in the group of rats receiving GBR 12909 alone, the time of walking was shortened at postnatal days p32, p62 and p92, but not at p44, compared to the control.

As to the second parameter, measured in OFT, i.e., the number of sector crossings, a two-way ANOVA also demonstrated a significant effect of BSO treatment in adolescent and adult rats ((for p44 (F_(1,36)_ = 8.510, *p* < 0.01); for p62 (F_(1,36)_ = 7.339, *p* < 0.001); for p92 (F_(1,36)_ = 32.777, *p* < 0.0001)) as well as interaction of BSO × GBR 12909 for two time points ((for p62 (F_(1,36)_ = 28.006, *p* < 0.001); for p92 (F_(1,36)_ = 18.836, *p* < 0.0001)). At none of the time points examined, a two-way ANOVA showed an impact of GBR 12909 treatment on the number of sector crossings. Post hoc analysis revealed that in rats administered BSO + GBR 12909 at postnatal days p5–p16, the number of sector crossings was significantly increased at p44, p62 and p92 when compared to groups treated with the vehicle (control), BSO or GBR 12909 alone (Figure 3B).

Regarding the third parameter measured in the OFT, i.e., the number of peeping and rearing episodes, a two-way ANOVA revealed a significant treatment effect of BSO ((for p44 (F_(1,36)_ = 5.586, *p* < 0.05); for p62 (F_(1,36)_ = 4.464, *p* < 0.05); for p92 (F_(1,36)_ = 24.027, *p* < 0.0001)) as well as an interaction of BSO × GBR 12909 ((for p44 (F_(1,36)_ = 14.004, *p* < 0.001); for p62 (F_(1,36)_ = 36.760, *p* < 0.001); for p92 (F_(1,36)_ = 27.376, *p* < 0.0001)). This analysis also showed a significant treatment effect of GBR 12909 for early adolescence (for p32 (F_(1,36)_ = 4.429, *p* < 0.05) and for adulthood (for p92 (F_(1,36)_ = 11.287, *p* < 0.002). Post hoc analysis demonstrated that at conditions of the combined administration of BSO + GBR 12909 at postnatal days p5–p16, the number of peeping and rearing episodes in adolescence (p44, p62) and adulthood (p92) was significantly higher than in groups that were treated with the vehicle (control), BSO or GBR 12909 alone during early postnatal life (Figure 3C).

### 2.4. The Impact of Chronic Administration of BSO and GBR 12909 during the Early Postnatal Life on the Spontaneous and AMF-Induced Locomotor Activity and Stereotypy in Adult Rats Determined Using Actometers

In order to find an association between GSH deficiency in early postnatal development (p5–p16) and hyperfunction of dopaminergic transmission and schizophrenia-like positive symptoms in adulthood, AMF at a single dose of 1 mg/kg was given to 90-day-old Sprague–Dawley adults previously treated with vehicle, BSO, GBR 12909 or BSO + GBR 12909. The effects of this DA-releasing stimulant on the total horizontal and vertical locomotor activity and on stereotypical behavior were compared with the effects of single saline injections in analogous groups of rats (Figure 4).

A two-way ANOVA performed for the total distance traveled in groups of rats receiving in adulthood a single dose of saline (Figure 4A) showed a significant treatment effect of BSO (F_(1,28)_ = 11.268, *p* < 0.01), but a lack of treatment effect of GBR 12909 (F_(1,28)_ = 3.360, NS) and no interaction between these two compounds (F_(1,28)_ = 1.031, NS). However, the same analysis carried out for this parameter in rats receiving a single dose of AMF in adulthood, revealed significant treatment effects both of BSO (F_(1,28)_ = 48.068, *p* < 0.0001) and GBR 12909 (F_(1,28)_ = 20.797, *p* < 0.0001) as well as an interaction between these model compounds (F_(1,28)_ = 19.459, *p* < 0.0001). Post hoc comparison of spontaneous horizontal locomotor activity, defined as the total distanced traveled, in four groups of rats treated with model compounds at postnatal days p5–p16 and receiving single doses of saline in adulthood, showed that this parameter increased significantly, only in the group receiving BSO + GBR 12909 combination when compared to the control and GBR 12909-treated group. The same comparison performed for rats receiving single doses of AMF in adulthood (Figure 4A) showed that the total distances traveled in all these groups were significantly longer than in the corresponding saline-treated rats. However, the most pronounced increase in the value of this parameter was observed in the BSO + GBR 12909 treated group receiving a single dose of AMF when compared to the same group of rats injected with saline. The latter effect means that AMF exacerbates the expression of locomotor activity, particularly in the BSO + GBR-treated group of rats.

As to the vertical locomotor activity, i.e., the total time spent on climbing, a two-way ANOVA showed only a significant treatment effect of BSO for a set of groups receiving either a single injection of saline (F_(1,28)_ = 6.218, *p* < 0.02) or AMF (F_(1,28)_ = 10.457, *p* < 0.01) in adulthood. Post hoc analysis of the total time spent on climbing in four groups of rats receiving a single dose of saline in adulthood showed that only in the group treated with BSO + GBR the value of this parameter was significantly higher than in the control and groups receiving only BSO or GBR 12909 (Figure 4B). A single dose of AMF increased the climbing time in all studied groups, but the strongest effect was found in the group treated with the BSO + GBR combination (Figure 4B). The latter effect clearly indicates that AMF, in addition to horizontal activity, also intensified vertical locomotor activity in the BSO + GBR-treated group.

A two-way ANOVA performed for the time of stereotypical behavior, expressed in seconds, in groups of rats receiving a single dose of saline in adulthood (Figure 4C), revealed only a significant treatment effect of GBR 12909 (F_(1,28)_ = 6.284, *p* < 0.05), but a lack of treatment effect of BSO (F_(1,28)_ = 0.575, NS) and no interaction between these two compounds (F_(1,28)_ = 0.001, NS). However, a post hoc analysis did not show any differences in the values of this parameter between the studied groups (Figure 4C). As to the time of stereotypical in four groups of rats receiving a single dose of AMF in adulthood, a two-way ANOVA demonstrated neither treatment effects of BSO (F_(1,28)_ = 1.622, NS) or GBR 12909 (F_(1,28)_ = 0.551, NS) alone nor an interaction between these two model compounds (F_(1,28)_ = 0.711, NS). However, comparisons using the Student’s t-test for independent samples, regarding the time of stereotypic behavior between the corresponding controls, BSO-, GBR 12909 and BSO + GBR 12909 groups administered a single dose of saline or AMF during adulthood, showed that AMF increased the duration of stereotypy in all studied groups (Figure 4C).

### 2.5. The Impact of Chronic Administration of BSO and GBR 12909 during the Early Postnatal Life on BDNF mRNA and Protein Levels in the Prefrontal Cortex and Hippocampus of Adult Rats

A two-way ANOVA performed for BDNF mRNA expression in the prefrontal cortex of adult rats receiving saline, BSO, GBR 12909 or BSO + GBR 12909 during early postnatal life revealed a significant treatment effect of BSO (F_(1,28)_ = 11.806, *p* < 0.002), a lack of treatment effect of GBR 12909 (F_(1,28)_ = 3.5876, *p* = 0.07) and an interaction between these compounds (F_(1,28)_ = 7.405, *p* < 0.01). The same analysis performed for hippocampal samples demonstrated only a significant interaction between BSO and GBR 12909 (F_(1,28)_ = 22.516, *p* < 0.0001), but no treatment effects of these model compounds when administered alone (for BSO F_(1,28)_ = 0.220, NS; for GBR 12909 F_(1,28)_ = 0.387, NS) (Figure 5).

In the PFC, a post hoc comparison showed that the levels of BDNF mRNA in BSO-, GBR 12909, and BSO + GBR-treated groups were significantly lower than in the control (Figure 5A). In the HIP, BDNF mRNA levels in the groups of rats administered only BSO or GBR 12909 alone were also lower than in the control group, but after the combined treatment (BSO + GBR 12909), the content of BDNF mRNA was significantly higher than in groups receiving model compounds separately and was almost the same as in the control (Figure 5C).

Regarding BDNF protein levels in the PFCs of adult rats chronically administered saline, BSO, GBR 12909 or BSO + GBR 12909 early in postnatal life, a two-way ANOVA showed a significant treatment effect of BSO (F_(1,24)_ = 6.745, *p* < 0.05) but no effect of GBR 12909 (F_(1,24)_ = 1.876, NS) as well as no interaction between these model compounds (F_(1,24)_ = 2.732, NS). The same analysis performed for the BDNF protein in the HIP revealed a significant effect of BSO treatment (F_(1,24)_ = 60.719, *p* < 0.0001) a lack of treatment effect of GBR 12909 (F_(1,24)_ = 2.817, NS) and a significant interaction between these compounds (F_(1,24)_ = 14.716, *p* < 0.001).

Post hoc comparisons of the studied groups demonstrated that the levels of BDNF protein in the PFC of adult rats receiving during BSO, GBR, or the combination of BSO + GBR 12909 postnatal days 5–16 were significantly reduced vs. the control (Figure 5B). The same comparisons performed in the HIP showed that BSO alone or in combination with GBR 12909 significantly reduced while GBR 12909 alone increased the levels of BDNF protein in this structure (Figure 5D).

## 3. Discussion

In the present study, a comprehensive analysis of the time-dependent onset of schizophrenic-like symptoms in Sprague–Dawley rats in which both GSH synthesis and dopamine (DA) reuptake were inhibited by BSO and GDR 12909, respectively, during the early postnatal life, was performed based on behavioral tests (SIT, NOR, OFT). The long-term effects of these model substances administered alone or in combination clearly showed that the first deficits in social behavior and cognitive functions appeared in mid-puberty (p42–p43) and were still present in adulthood (p90–p91) both in the group of rats receiving BSO alone or a combination of BSO + GBR 12909. However, only rats treated with BSO + GBR 12909 showed the increased exploratory behavior assessed in the OFT as the time of walking, the number of sector crossings (ambulation) as well as the number of peeping and rearing episodes that are considered to be equivalent to positive symptoms in patients with schizophrenia. They appeared for the first time on the postnatal day p44 and continued to manifest into adulthood (p92). Unlike the groups treated with BSO alone or with the BSO + GBR 12909 combination, rats receiving GBR 12909 alone in early postnatal life showed no deficits in social behavior, as measured by the total interaction time and the number of interactions, both during adolescence and early adulthood. Regarding cognitive functions in rats treated with GBR 12909, a significant transient decrease in the value of a recognition index, which is a measure of these functions, was only observed on p43, followed by recovery of the recognition index value to the control group level on days p61 and p91.

In order to introduce the principles of cognitive function assessment based on the NOR test, it should be reminded that this test is based on the spontaneous tendency of rats to investigate objects and to favor novel objects versus familiar ones. In our study, rats receiving BSO alone early in postnatal life failed to discriminate between familiar and novel objects at all time points studied. Since basal exploratory activity, as assessed by the OFT, was preserved in this group of rats at virtually all time points, significant decreases in the value of recognition index during adolescence (p43, p61) and early adulthood (p92) can be attributed to cognitive impairment.

In contrast to the BSO-treated group, rats administered GBR 12909 alone early in the postnatal life examined the novel object much less intensively than the familiar one only in the middle adolescence (p43). However, as the basal exploratory activity measured in the OFT remained at the level of the control group, therefore, a significant decrease in the recognition index value at p43 can also be attributed to cognitive impairment. Furthermore, during subsequent adolescence and adulthood, the interest of the GBR 12909-treated rats in the investigation of the novel object gradually increased. This increased interest in studying the novel object compared to the old one was the greatest on p91 and occurred despite the significant reduction in rat exploratory activity observed in the OFT. The latter behavioral data under discussion indicate that a transient cognitive decline on p43, induced by disturbances only in the dopaminergic transmission during early postnatal life, may be gradually compensated in late adolescence (p61), eventually reaching cognitive normalization in early adulthood (p91).

In contrast, rats receiving the BSO + GBR12909 combination in early postnatal life, like those receiving BSO alone, were unable to distinguish between familiar and novel objects in the NOR test performed in adolescence and adulthood. However, the exploratory activity measured in the OFT as the time of walking was significantly increased in rats receiving the BSO + GBR combination compared to the control group and those receiving BSO- or GBR 12909 alone. Thus, the presence of cognitive impairment, assessed in the group of rats receiving the BSO + GBR 12909 combination based on the recognition index in adolescence and early adulthood, may be a consequence of the increased dopaminergic transmission that occurred under conditions of inhibited GSH synthesis in the early postnatal life. The mechanism and the underlying neuronal basis of the BSO + GBR 12909-mediated cognitive impairment remains to be elucidated. In conclusion, the observed disturbances in the NOR test in rats treated with BSO alone or BSO + GBR combination are in line with the decreased object recognition capacity of schizophrenic patients as compared to healthy control subjects [64,65,66].

Simultaneously with the occurrence of cognitive disturbances, in the group of rats treated with BSO + GBR 12909, behaviors corresponding to positive symptoms observed in patients with schizophrenia were also found. The presence of positive symptoms of schizophrenia in this group of rats was confirmed in the previously described OFT, and in an additional experiment carried out on 91-day-old adult rats, in which the horizontal (distance travel) and vertical (climbing) locomotor activities were measured using actometers as equivalents of these symptoms. These motor activity parameters in the BSO + GBR 12909-treated rats were significantly enhanced compared to groups treated with saline, BSO, or GBR 12909 alone, respectively, in the early postnatal life. However, the most important effect was that only in the group of rats receiving the BSO + GBR 12909 combination, AMF given in adulthood exacerbated the positive symptoms compared to the corresponding group of rats receiving saline instead of AMF. This AMF-mediated exacerbation of already elevated motor parameters in rats treated with the BSO + GBR 12909 combination clearly indicates that in this neurodevelopmental rat model of schizophrenia, like in schizophrenia patients [43,44,45], AMF may worsen the existing positive symptoms. These behavioral data also show that Sprague–Dawley rats treated with the combination BSO + GBR 12909 at early postnatal days (p5–p16) reproduce virtually all the symptoms seen in schizophrenia patients and can, therefore, be considered a valuable neurodevelopmental model of schizophrenia for studying the efficacy of antipsychotic drugs.

Since BDNF has an established role in neuronal development and synaptogenesis [50,67] and is an important modulator of monoaminergic and GABA-ergic neurotransmitter systems [68], in groups of rats chronically treated with BSO and GBR 12909 alone or in combination in the early postnatal period, we decided to test the long-term effects of this treatment on BDNF mRNA and its protein levels in the PFC and HIP in adulthood. Our results show that in the PFC, administration of BSO alone or the combination of BSO + GBR 12909 resulted in decreases in both BDNF mRNA and protein levels in adulthood. In the HIP, in the group of rats receiving only BSO, just as in the PFC, the decrease in the level of BDNF mRNA was accompanied by a decrease in its protein content. However, in the HIP of BSO + GBR-treated rats, despite no changes in the level of BDNF mRNA, a significant decrease in BDNF protein level was observed both when compared to the control and GBR-12909-treated groups. Interestingly, in the group of rats receiving GBR 12909 alone in early postnatal life, in which no social and cognitive deficits were revealed in adulthood, the levels of BDNF mRNA and its protein in the PFC were significantly reduced, while in the HIP despite a significant decrease in BDNF mRNA a relatively large increase in BDNF protein was observed compared to the control group and that treated with BSO alone. The latter results show that GBR 12909 administered alone in the early postnatal period modulates BDNF protein expression in adulthood in an inverse manner, reducing its level in the PFC and simultaneously increasing it in the HIP. In addition, these results suggest that in the group receiving GBR 12909 alone, the increase in the BDNF protein level in the HIP may be of compensatory nature, ultimately leading to the normalization of cognitive functions. However, such a compensatory effect at the BDNF protein level did not occur in the rat HIP when BSO and GBR 12909 were administered in combination. Our data suggest that redox imbalance [14,62,69,70], as a result of repeated treatment with a GSH synthesis inhibitor (BSO) administered alone or in combination with a DA reuptake inhibitor (GBR 12909) during early postnatal development, may be an important factor reducing the expression of BDNF protein in the PFC and HIP in adulthood. In addition, in another model of schizophrenia [71] in rats with a neonatal ibotenic lesion of the ventral HIP, a reduction in the level of BDNF mRNA in the PFC and HIP was demonstrated [72]. In general, in other animal studies, it was shown that some early life events could produce long-lasting effects on processes to which neurotrophins contribute, thereby affecting neuronal maturation and plasticity in later periods of life [68,73,74].

According to the dopamine hypothesis of schizophrenia, it is postulated that the hypofunction of the cortical and prefrontal dopamine systems contributes to negative symptoms and cognitive deficits and that the subcortical and limbic dopamine system hyperactivity causes positive symptoms of schizophrenia [75,76,77]. In our study, it is difficult to explain the appearance of social and cognitive deficits in adolescence and early adulthood in the group of rats receiving BSO alone during the early postnatal life, referring only to the above-mentioned dopamine hypothesis of schizophrenia. These data show that their occurrence in the BSO-treated group may be related to disturbances in other neurotransmitter systems rather than the dopamine system [78]. Since the inhibition of GSH synthesis may impact the function of NMDA receptors [14,62,69], it seems that impairment of the excitation-inhibition balance during the early postnatal life plays a decisive role in the occurrence of social and cognitive deficits in the BSO-treated rats later in life. Our neurodevelopmental rat model of schizophrenia induced by chronic BSO treatment in the early postnatal days seems to be consistent with the NMDA receptor insufficiency model of schizophrenia postulated previously by Carlsson [79]. To validate the rat model of schizophrenia presented here in screening drugs that can be used in the therapy, we recently showed that chronic per os administration of the antioxidant *N*-acetylcysteine at a dose of 30 mg/kg to adult rats treated with BSO resulted in the reversal of social and cognitive deficits assessed by the SIT and NOR tests (data in preparation). In this model, antipsychotic drug aripiprazole administered chronically (0.3 and 1 mg/kg i.p.) was also effective in reversing social and cognitive deficits in adulthood [80]. Although in the BSO-treated rats it is difficult to determine the role of dopaminergic transmission in the development of social and cognitive deficits in adulthood, in the group of rats receiving the BSO + GBR 12909 combination in early postnatal life, the contribution of in the dopaminergic transmission in shaping them should be seriously considered especially because the changes in rats’ behavior corresponding to positive symptoms in patients with schizophrenia were observed only in the latter group.

Furthermore, only in this group of rats, AMF exacerbated the existing positive symptoms. The dopamine hypothesis of schizophrenia has been linked to hippocampal hyperactivity. Dopamine released in the hippocampus improves the performance of some specific cognitive tasks at higher concentrations but has detrimental effects at levels above the optimal range [81,82]. The effect induced by AMF in the BSO + GBR 12909-treated group is consistent with the above-presented data.

## 4. Conclusions

Our data show that redox status disturbances caused by chronic treatment with BSO alone during early postnatal life lead to social and cognitive deficits that appeared in the middle adolescence stage and continued to occur into adulthood, as well as to the decreases in BDNF mRNA and its protein levels in the PFC and HIP in adulthood. When redox status disturbances were accompanied by disruption of dopaminergic transmission induced by GBR 12909 treatment during early postnatal life, in addition to social and cognitive deficits, rats were shown to develop behaviors corresponding to the positive symptoms in humans. In the latter case, the decreases in BDNF mRNA and its protein levels were observed only in the PFC, but in the HIP, only the level of BDNF protein was decreased. These results suggest a causal relationship between BDNF deficiency and the occurrence of schizophrenia-like symptoms in this neurodevelopmental rat model of schizophrenia.

## 5. Materials and Methods

The experiments were carried out in compliance with the Act on Experiments on Animals of 21 January, 2005 reapproved on 15 January, 2015 (published in Journal of Laws no 23/2015 item 266, Poland), and according to the Directive of the European Parliament and of the Council of Europe 2010/63/EU of 22 September 2010 on the protection of animals used for scientific purposes. The studies received also an approval of the Local Ethics Committee at the Institute of Pharmacology, Polish Academy of Sciences (permission no 3/2018 of January 2018). All efforts were made to minimize the number and suffering of animals used.

### 5.1. Animals and Treatment

To create the neurodevelopmental model of schizophrenia, pregnant Sprague–Dawley females at embryonic day 16 were delivered to our laboratory by the Charles River Company (Sulzfeld, Germany). They were kept in individual cages under standard laboratory conditions, at room temperature (22 °C) under an artificial light/dark cycle (12/12 h), with free access to standard laboratory food and tap water. On the day of parturition, the sex of pups was determined, and only males were left with their mother to be used in the further experimental procedure. Between the postnatal days p5 and p16, male Sprague–Dawley pups were administered the selective inhibitor of GCL, compound l-buthionine-(*S*,*R*)-sulfoximine (BSO, 3.8 mmol/kg s.c., once daily), and the dopamine reuptake inhibitor 1-[2-Bis(4-fluorophenyl)methoxy]ethyl]-4-(3-phenylpropyl)piperazine dihydrochloride (GBR 12909,5 mg/kg s.c., every second day), alone or in combination. At conditions of the combined treatment, BSO administration preceded an injection of GBR 12909. Control rats, instead of BSO or GBR 12909, received a vehicle once daily. Rats were weighed daily, and the injected volumes of the studied model compounds were adjusted accordingly to the actual body weight. On postnatal day p23, rats were weaned and housed in groups of four to five until p92. Behavioral tests assessing the expression of schizophrenia-like changes, corresponding to negative (social interaction test; SIT) and positive (open field test; OFT) symptoms, and to cognitive deficits (new object recognition test; NOR), were carried out successively in all groups of rats, in the early (p30), middle (p42–p44) and late adolescence (p60–p62) as well as in adulthood (p90–p92). In additional groups of rats receiving vehicle (control), BSO-, GBR 12909- and BSO + GBR 12909 at postnatal days (p5–p16), spontaneous and amphetamine-induced locomotor activity (horizontal and vertical) and stereotypic behavioral patterns were measured with actometers in adulthood (on the 90th day). d-amphetamine (AMF) was administered at a single dose of 1 mg/kg, s.c. and after 30 min, the measurement was initiated and lasted 30 min.

### 5.2. Social Interaction Test

The social interaction test (SIT) was performed using a black PCV box (67 × 57 × 30 cm, length × width × height). The arena was dimly illuminated with an indirect light of 18 Lux [83]. Each social interaction experiment involving two rats was carried out during the light phase of the light/dark cycle. The rats were selected from separate housing cages to make a pair for the study. The paired rats were matched for body weight within 15 g. Each pair of rats was diagonally placed in opposite corners of the box facing away from each other. The behavior of the animals was measured over a 10-min period. The test box was wiped clean between each trial. Social interaction between two rats was expressed as the total time spent in social behavior, such as sniffing, genital investigation, chasing, and fighting with each other. The number of episodes was counted as a separate paradigm. The SIT was performed 4 times: on days p30, p42, p60, and p90. Each group was composed of 16 rats (8 pairs).

### 5.3. Novel Object Recognition Test

The novel object recognition (NOR) test was performed using a black PCV box (67 × 57 × 30 cm, length × width × height). The arena was dimly illuminated with an indirect light of 18 Lux. On the first day of the experiment (adaptation), rats were placed in the box for 10 min. On the next day, the animals were placed in the box for 5 min (T1) with two identical objects (white tin 5 cm wide and 14 cm high or green pyramid 5 cm wide and 14 cm high). The time of object exploration was measured for each of the two objects separately. Then, one hour after T1, the rats again were placed on the box for 5 min (T2), with two different objects: one from the previous session (old) and the other new (white box and green pyramid). The time of object exploration was measured for each of the two objects separately (sniffing, touching, or climbing). NOR test was performed 4 times: on days p31, p43, p61, and p91. Each group was composed of 10 rats.

### 5.4. Open Field Test

Exploratory activity was assessed in the elevated open field test (OFT). A black circular platform without walls having 1 m in diameter was divided into six symmetrical sectors and was elevated 50 cm above the floor. The laboratory room was dark, and only the center of the open field was illuminated with a 75 W bulb placed 75 cm above the platform. At the beginning of the test, the animal was placed gently in the center of the platform and was allowed to explore. The exploratory activity, ambulation, peeping, and rearing in the open field, i.e., respectively, the time of walking, the number of sector crossings, and the number of episodes of peeping under the edge of the arena and rearing were assessed for 3 min. The OFT was performed 4 times: on days p32, p44, p63, and p93. Each group consisted of 10 rats.

### 5.5. Locomotor Activity Assessed in Actometers

Spontaneous and AMF-induced locomotor activities of adult 90-day-old Sprague–Dawley rats were recorded individually for each animal using the Opto-Varimex cages (Columbus Instruments, Columbus, OH, USA) linked online to a compatible IBM-PC. Each cage (43 × 44 × 25 cm) was surrounded with a 15 × 15 array of photocell beams located 3 cm from the floor surface as previously. Interruptions of the photocell beams were used to measure horizontal locomotor activity, defined as a distance traveled and expressed in cm, while the time of vertical locomotor activity (climbing) or the time of stereotypy was expressed in seconds. These three parameters were measured for 30 min, starting 30 min after AMF (1 mg/kg, s.c.) administration.

### 5.6. BDNF Expression Analysis

Freshly isolated rat frontal cortex and hippocampus tissues were stored at −80 °C prior to the next analysis. Total RNA was isolated using commercially available Bead-Beat Total RNA Mini Kit (A&A Biotechnology, Gdansk, PL) according to the manufacturer’s instructions. The concentration of total RNA was determined spectrophotometrically by using NanoDrop (ND/1000 UV/Vis, Thermo Fisher NanoDrop, Waltham, MA, USA). After dissolving in water, RNA (1 μg) was reverse-transcribed to cDNA using High Capacity cDNA Reverse Transcription kit with RNAse inhibitor and random hexamers (MultiScribe™, Applied Biosystems, Life Technologies, Carlsbad, CA, USA). cDNA was synthesized by using thermal cycler T100TM Thermal Cycler (Bio-Rad Laboratories, Hercules, CA, USA) in RT-life program conditions: 10 min at 25 °C, 120 min at 37 °C, 5 min at 85 °C, and indefinitely at 4 °C.

The BDNF mRNA level was determined by real-time PCR using predesigned TaqMan Gene Expression Assays (Applied Biosystems, Thermo Fisher Scientific, Milton Keynes, UK). Assay IDs for the genes examined were as follows: BDNF (Rn01484925_m1) and for reference’s gene HPRT1 (Rn01527840_m1). Amplification was carried out in a total volume of 10 μL (FCx). The mixture containing: 1 × FastStart Universal Probe Master (Rox) mix (Roche, Germany), 900 nM TaqMan forward and reverse primers, and 250 nM of hydrolysis probe labeled with the fluorescent reporter dye FAM at the 5′-end and a quenching dye at the 3′-end and RNAse free water. We used 50 ng of cDNA for the PCR template, real-time PCR was conducted using a thermal cycler Quant Studio 3 (Thermo Fisher Scientific, Waltham, MA, USA), and thermal cycling conditions were: 2 min at 50 °C and 10 min at 95 °C followed by 40 cycles at 95 °C for 15 s and at 60 °C for 1 min. Samples were run in duplicate.

### 5.7. ELISA Assay

Freshly isolated rat frontal cortex and hippocampus tissues were stored at −80 °C prior to the analysis. First, the tissues were rinsed in DPBS (GIBCO, Thermo Fisher Scientific, Waltham, MA, USA) to remove excess blood, and then they were homogenized in DPBS using Tissue Lyser II (Qiagen Inc., Valencia, CA, USA). The protein measurements of all samples were performed using a BCA Protein Assay Kit (Sigma-Aldrich, St. Louis, MO, USA) in pursuance of the manufacturers’ instructions. The protein contents were assessed using a Tecan Infinite 200 Pro spectrophotometer (Tecan, Mannedorf, Germany). Samples containing hippocampal and cortical supernatants were analyzed by enzyme-linked immunoassay (ELISA) using commercially available kits: Rat Brain-Derived Neurotrophic Factor ELISA Kit, cat.no. E0476Ra (Bioassay Technology Laboratory, Shanghai, China) according to the manufacturers’ instructions. Briefly, 50 µL of standards and 40 µL of samples, respectively, were dispensed into 96-well coated plates. Next, 10 µL of anti-BDNF antibodies were added into wells containing samples, and 50 µL streptavidin-HRP was added to each well except for blank, and the samples were incubated for 60 min at 37 °C. After washing and the next steps as recommended by the manufacturer, the absorbance was determined using a Tecan Infinite 200 Pro spectrophotometer (Tecan, Mannedorf, Germany) set to 450 nm.

### 5.8. Statistics

The statistical analysis of the obtained behavioral data was performed using a two-way ANOVA followed (if significant) by the Newman–Keuls test for post hoc comparisons. In addition, for a comparison of two groups, the Student’s *t*-test for independent samples was used.

## Figures and Tables

**Figure 1 ijms-22-06171-f001:**
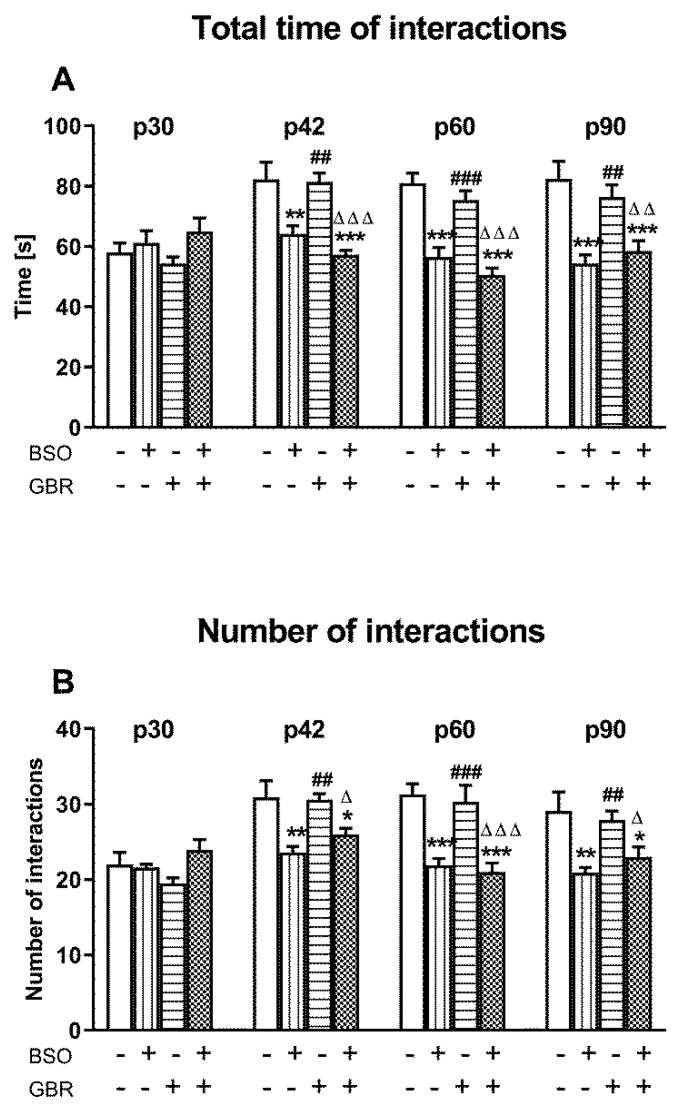
The effect of chronic administration of BSO and GBR 12909, alone or in combination, during postnatal days p5–p16 on the social behavior assessed as the total time spent in social interactions (**A**) and the number of interactions (**B**) in adolescent and adult Sprague–Dawley rats. Data are presented as the mean ± SEM, *n* = 16 (8 pairs) for each group. Statistical analysis was performed using a two-way ANOVA; symbols indicate significance of differences according to the Newman–Keuls post hoc test, *** *p* < 0.001, ** *p* < 0.01, * *p* < 0.05 vs. control; ^###^ *p* < 0.001, ^##^ *p* < 0.01 vs. BSO- and ^∆∆∆^ *p* < 0.001, ^∆∆^ *p* < 0.01, ^∆^ *p* < 0.05 vs. GBR 12909-treated groups.

**Figure 2 ijms-22-06171-f002:**
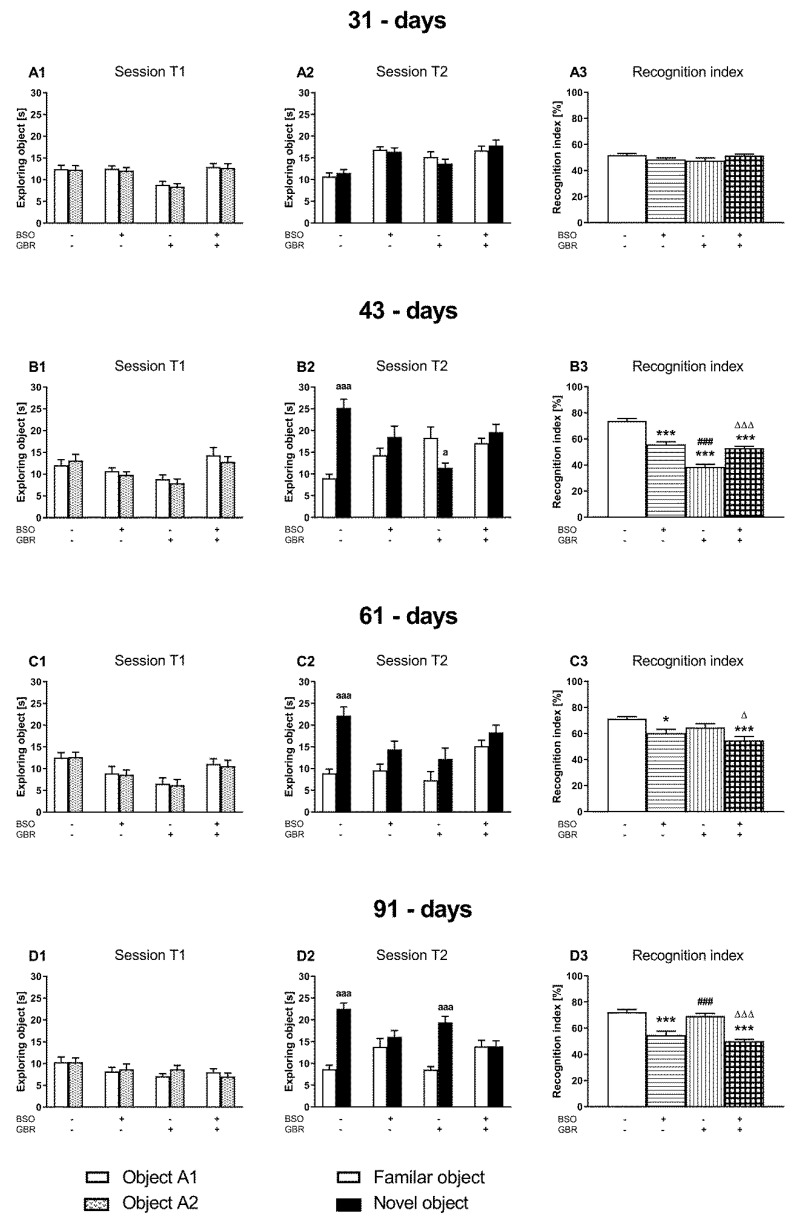
The effect of chronic administration of BSO and GBR 12909, alone or in combination, during postnatal days p5–p16 on cognitive functions assessed in adolescent (**A**–**C**) and adult (**D**) Sprague–Dawley rats. (**A1**–**D1**) The effects of the studied model compounds on the exploration of two identical objects in the acquisition trials (session T1). (**A2**–**D2**) The effects of the studied model compounds on the exploration of a novel and familiar object in the retention trial (Session T2). (**A3**–**D3**) The effects of the studied model compounds on the recognition index. Data are presented as the mean ± SEM, *n* = 10 for each group. Letters indicate statistically significant differences between the exploration time of a novel and familiar object in the session T2 within each studied group, according to the Student’s *t*-test for independent samples, ^aaa^ *p* < 0.001, ^a^ *p* < 0.05 vs. familiar object. Statistical analysis of the recognition index was performed using a two-way ANOVA; symbols indicate significance of differences according to the Newman–Keuls post hoc test, *** *p* < 0.001, * *p* < 0.05 vs. control; ^###^ *p* < 0.001 vs. BSO- and ^∆∆∆^ *p* < 0.001, ^∆^ *p* < 0.05 vs. GBR 12909-treated groups.

**Figure 3 ijms-22-06171-f003:**
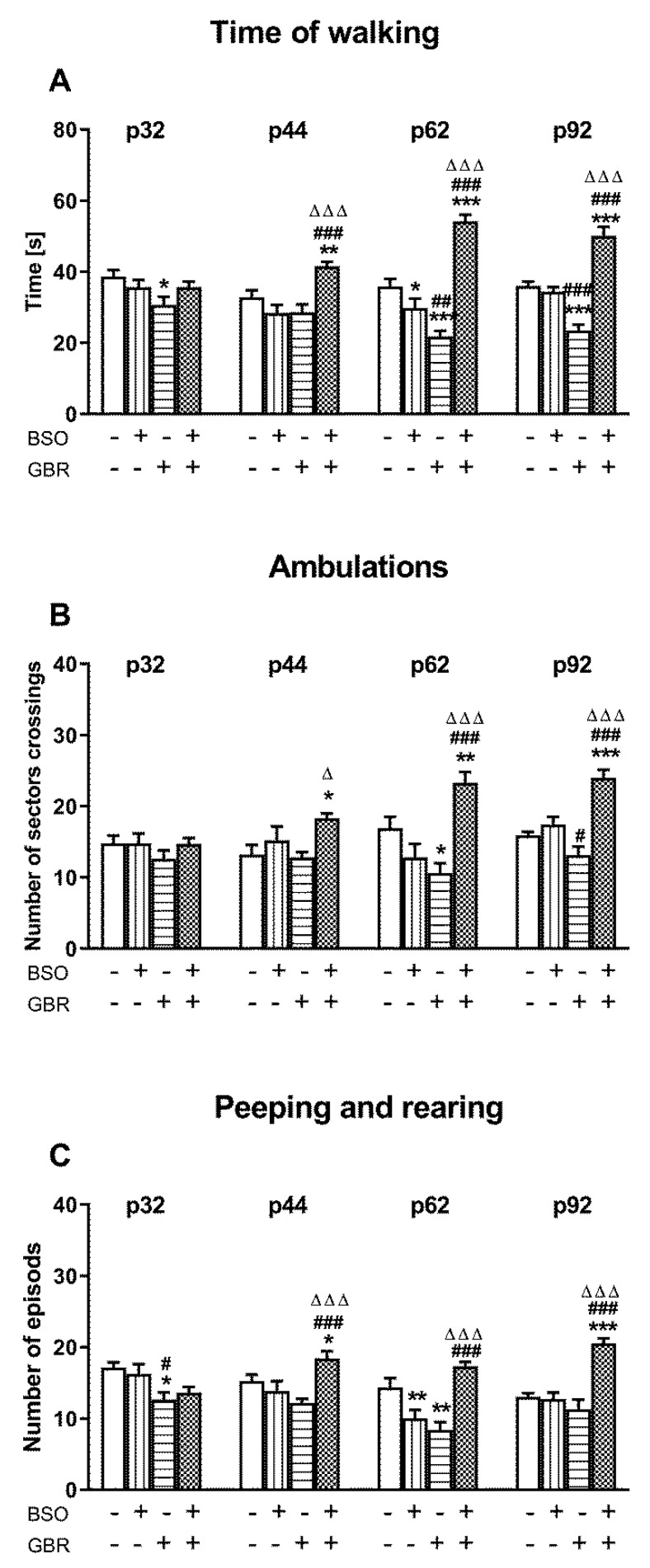
The effect of chronic administration of BSO and GBR 12909, alone or in combination, during postnatal days p5–p16 on positive symptoms assessed in the OFT in adolescent and adult Sprague–Dawley rats as: (**A**) the time of walking (**B**) the number of sector crossings (**C**) the number of peeping and rearing episodes. Data are presented as the mean ± SEM, *n* = 10 for each group. Statistical analysis was performed using a two-way ANOVA; symbols indicate significance of differences according to the Newman–Keuls post hoc test, *** *p* < 0.001, ** *p* < 0.01, * *p* < 0.05 vs. control; ^###^ *p* < 0.001, ^##^ *p* < 0.01, ^#^ *p* < 0.05 vs. BSO and ^∆∆∆^ *p* < 0.001, ^∆^
*p* < 0.05 vs. GBR 12909-treated groups.

**Figure 4 ijms-22-06171-f004:**
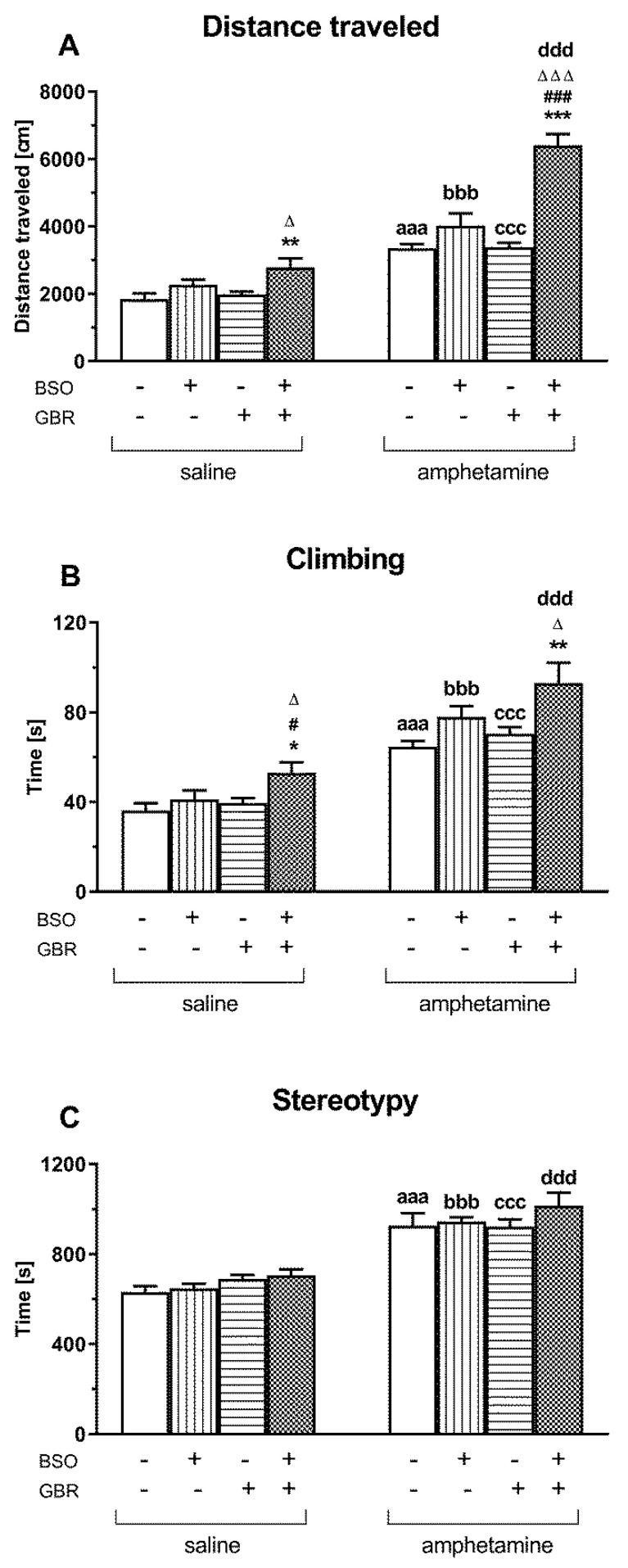
The effect of chronic administration of BSO and GBR 12909, alone or in combination, during postnatal days p5–p16, on the spontaneous and AMF-induced locomotor activity and stereotypy, measured in 90-day-old Sprague–Dawley rats using actometers. (**A**) Horizontal locomotor activity is presented as the total distance traveled expressed in cm, (**B**) vertical locomotor activity is shown as the total time spent climbing expressed in seconds (s), and (**C**) stereotypy as the total time devoted to stereotypical behavior expressed in seconds. These parameters were recorded during a 30–minute measurement session. Data are presented as the mean ± SEM, *n* = 10 for each group. Statistical analysis was performed using a two-way ANOVA; symbols indicate significance of differences according to the Newman–Keuls post hoc test, *** *p* < 0.001, ** *p* < 0.01, * *p* < 0.05 vs. control; ^###^ *p* < 0.001, ^#^ *p* < 0.05 vs. BSO-; and ^∆∆∆^ *p* < 0.001, ^∆^ *p* < 0.05 vs. GBR 12909-treated groups. Comparisons between the corresponding groups treated with saline or AMF were performed using the Student’s *t*-test for independent samples, ^aaa^ *p* < 0.001 vs. saline-treated control, ^bbb^ *p* < 0.001 vs. saline-treated BSO group; ^ccc^ *p* < 0.001 vs. saline-treated GBR 12909 group; ^ddd^ *p* < 0.001 vs. saline-treated BSO + GBR 12909 group.

**Figure 5 ijms-22-06171-f005:**
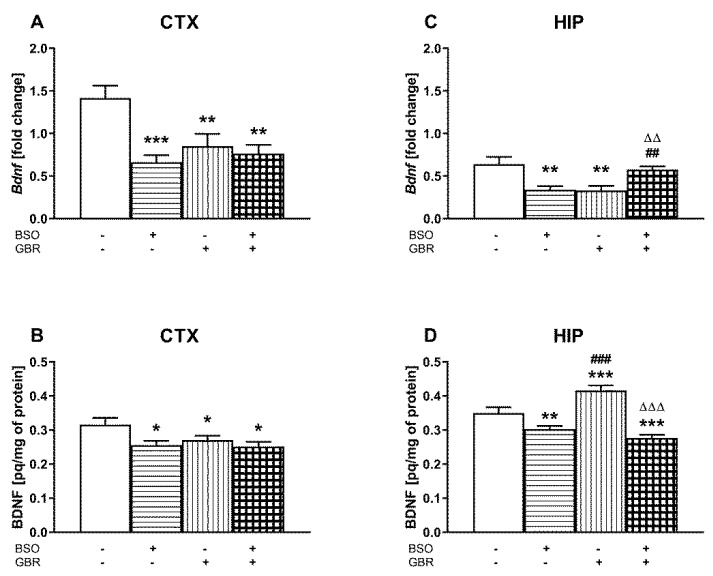
The effect of chronic administration of BSO and GBR 12909, alone or in combination, during postnatal days p5–p16, on BDNF mRNA and protein levels in the PFC (**A**,**C**) and HIP (**B**,**D**) of adult rats. Data are presented as the mean ± SEM, *n* = 7–8 for each group. Statistical analysis was performed using a two-way ANOVA; symbols indicate significance of differences according to the Newman–Keuls post hoc test, *** *p* < 0.001, ** *p* < 0.01, * *p* < 0.05 vs. control; ^###^ *p* < 0.001, ^##^ *p* < 0.01, vs. BSO-; and ^∆∆∆^ *p* < 0.001, ^∆∆^ *p* < 0.01 vs. GBR 12909-treated groups.

## Data Availability

Data supporting reported results are available on request from the corresponding author.

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
