# Peer review of "Glutathione Deficiency during Early Postnatal Development Causes Schizophrenia-Like Symptoms and a Reduction in BDNF Levels in the Cortex and Hippocampus of Adult Sprague–Dawley Rats"

_ijms, 2021, doi:10.3390/ijms22126171_

Round 1

Reviewer 1 Report

The paper by Lech and colleagues reports interesting results regarding the effects of BSO, a glutathione synthesis inhibitor, and GBR 12909, a dopamine reuptake inhibitor, given alone or in combination, on the development of schizophrenia-like behaviours in Sprague-Dawley rats, suggesting that this model could be considered as a valuable neurodevelopmental model of schizophrenia. The authors also examined the effect of BSO and GBR 12909 treatment on the expression of BDNF mRNA and its protein in the prefrontal cortex and hippocampus during adulthood.

The paper is well written and it needs only some minor language corrections. The authors should pay attention to the use of commas. Namely, many sentences are difficult to follow precisely because of the lack of commas that should serve to separate certain parts of the sentences.

Statistical analysis has been carefully done with all information given in methods and results. However, I have few comments and advices for ameliorating the paper.

Results:

Line 198: Please replace “animals” with “anomals” in the next sentence: “In the retention trial (session T2) adolescent control 43- and 61-day-old, but not 31-day-old ANIMALS…”

Line 257: Please rephrase the sentence: “At none of the time points examined, DID a two-way ANOVA SHOW an impact of GBR 12909 treatment on the number of sector crossings.”

Materials and methods:

Line 588: Please replace “was” with “where” in the next sentence: “On the next day, the animals WERE placed in the box for 5 minutes (T1) with two identical objects…”

BDNF expression analysis: Please give some more details regarding the RT-PCR reaction (similar as you did for the Real-Time PCR).

Author Response

Answer for the Reviewer 1

General comments

The paper by Lech and colleagues reports interesting results regarding the effects of BSO, a glutathione synthesis inhibitor, and GBR 12909, a dopamine reuptake inhibitor, given alone or in combination, on the development of schizophrenia-like behaviours in Sprague-Dawley rats, suggesting that this model could be considered as a valuable neurodevelopmental model of schizophrenia. The authors also examined the effect of BSO and GBR 12909 treatment on the expression of BDNF mRNA and its protein in the prefrontal cortex and hippocampus during adulthood.

The paper is well written and it needs only some minor language corrections. The authors should pay attention to the use of commas. Namely, many sentences are difficult to follow precisely because of the lack of commas that should serve to separate certain parts of the sentences.

Statistical analysis has been carefully done with all information given in methods and results. However, I have few comments and advices for ameliorating the paper.

Answer the general comments

The manuscript was checked by a professional translator, every effort has been made to make it more easily understandable.

Comments to the Results chapter

Line 198: Please replace “animals” with “anomals” in the next sentence: “In the retention trial (session T2) adolescent control 43- and 61-day-old, but not 31-day-old ANIMALS…”

Line 257: Please rephrase the sentence: “At none of the time points examined, DID a two-way ANOVA SHOW an impact of GBR 12909 treatment on the number of sector crossings.”

Reply to the comments in the Results chapter

In line 198, the word „anomals” has been replaced by „animals”.

In line 257, the sentence starting with the phrase „At none of the time points examined,…..has been corrected as suggested by the Reviewer 1.

Comments to the Material and Methods chapter

Materials and methods: Line 588: Please replace “was” with “where” in the next sentence: “On the next day, the animals WERE placed in the box for 5 minutes (T1) with two identical objects…”

BDNF expression analysis: Please give some more details regarding the RT-PCR reaction (similar as you did for the Real-Time PCR).

Reply to the comments in the Materials and Methods chapter

In line 588, the word „was” has been replaced by „were”.

BDNF expression analysis regarding the RT-PCR reaction was supplemented by two additional sentences (page…, lines….).

Reviewer 2 Report

The study is an example of a well-designed experimental study with direct clinical implications in the translational model. The experimental model developed by the authors refers to the currently accepted neurodevelopmental model of schizophrenia and (no described by the authors) the excessive production of free radicals as a consequence of dysfunctional NMDA receptors and disinhibition of transmission in neurotransmitter systems. The research model discovered by the researchers: inhibition of glutathione production, associated disturbances in BDNF production and schizophrenia-like symptoms monitoring create a new experimental axis in the neurodevelopmental model of schizophrenia and the possibility of studying the relationship between redox processes and the disease pathogenesis. Researchers are trying to relate their results to the dopaminergic hypothesis. In the meantime, a more recent model of schizophrenia has emerged, related to NMDA receptors insufficiency. Hypofunction of NMDA receptors on GABAergic interneurons leads to disturbances in the activity of the cortico-striatal-thalamo-cortical (CSTC) pathway, which results in excessive activity of the striatum. On the other hand, the NMDA receptors’ metaplasticity disturbances associated with the insufficiency of synaptic NMDA receptors lead to impaired neuroconectivity and to cognitive dysfunctions. The influence of glutathione on the metaplasticity of NMDA receptors may link the experimental model developed by the authors with the model of NMDA receptor insufficiency. This model was first described by Arvid Carlsson (doi: 10.1055 / s-2006-931483). In order to confirm this model, it is worth mentioning that a direct relationship has been demonstrated between the activity of antipsychotic drugs and the change in the structure of NMDA receptor subunits (doi: 10.3390 / ijms20061442). In this article, I miss references to possible clinical implications - the results may indicate a potential role of antioxidants in early intervention in the prevention of schizophrenia development. I suggest the authors try to confront their findings with the NMDA receptor insufficiency model in schizophrenia and try to present the potential clinical perspectives of their interesting findings.

Author Response

Answer for the Reviewer 2

Comments and Suggestions for Authors

The study is an example of a well-designed experimental study with direct clinical implications in the translational model. The experimental model developed by the authors refers to the currently accepted neurodevelopmental model of schizophrenia and (no described by the authors) the excessive production of free radicals as a consequence of dysfunctional NMDA receptors and disinhibition of transmission in neurotransmitter systems. The research model discovered by the researchers: inhibition of glutathione production, associated disturbances in BDNF production and schizophrenia-like symptoms monitoring create a new experimental axis in the neurodevelopmental model of schizophrenia and the possibility of studying the relationship between redox processes and the disease pathogenesis. Researchers are trying to relate their results to the dopaminergic hypothesis. In the meantime, a more recent model of schizophrenia has emerged, related to NMDA receptors insufficiency. Hypofunction of NMDA receptors on GABAergic interneurons leads to disturbances in the activity of the cortico-striatal-thalamo-cortical (CSTC) pathway, which results in excessive activity of the striatum. On the other hand, the NMDA receptors’ metaplasticity disturbances associated with the insufficiency of synaptic NMDA receptors lead to impaired neuroconectivity and to cognitive dysfunctions. The influence of glutathione on the metaplasticity of NMDA receptors may link the experimental model developed by the authors with the model of NMDA receptor insufficiency. This model was first described by Arvid Carlsson (doi: 10.1055 / s-2006-931483). In order to confirm this model, it is worth mentioning that a direct relationship has been demonstrated between the activity of antipsychotic drugs and the change in the structure of NMDA receptor subunits (doi: 10.3390 / ijms20061442). In this article, I miss references to possible clinical implications - the results may indicate a potential role of antioxidants in early intervention in the prevention of schizophrenia development. I suggest the authors try to confront their findings with the NMDA receptor insufficiency model in schizophrenia and try to present the potential clinical perspectives of their interesting findings.

Reply to the Comments and Suggestions for Authors

As suggested by the reviewer, to validate this neurodevelopmental model of schizophrenia in terms of clinical utility, the manuscript was supplemented with information from our researches currently being prepared for publication on the effects of the antipsychotic drug aripiprazole and N-acetylcysteine anoxidant in reversing social and cognitive deficits measured in the SIT and NOR tests.

This excerpt is in the penultimate paragraph of the discussion (page 16, lines509-523).

The introduced change was related to the addition of two items in the list of references (items 79, 80).
